# Rapid Detection of Total Viable Count in Intact Beef Dishes Based on NIR Hyperspectral Hybrid Model

**DOI:** 10.3390/s23239584

**Published:** 2023-12-03

**Authors:** Wensong Wei, Fengjuan Zhang, Fangting Fu, Shuo Sang, Zhen Qiao

**Affiliations:** 1Key Laboratory of Agricultural Product Processing, Ministry of Agriculture/Institute of Food Science and Technology, Chinese Academy of Agricultural Sciences, Beijing 100193, China; 2Zibo Institute for Digital Agriculture and Rural Research, Zibo 255051, China; zhangfengjuan0909@163.com (F.Z.); fufangting10@163.com (F.F.); sangshuo17@foxmail.com (S.S.); huashengqiao@163.com (Z.Q.)

**Keywords:** SVM-DS, crushed samples, spectral hybrid model, total viable count, freshness of dishes

## Abstract

The total viable count (TVC) of bacteria is an important index to evaluate the freshness and safety of dishes. To improve the accuracy and robustness of spectroscopic detection of total viable bacteria count in a complex system, a new method based on a near-infrared (NIR) hyperspectral hybrid model and Support Vector Machine (SVM) algorithms was developed to directly determine the total viable count in intact beef dish samples in this study. Diffuse reflectance data of intact and crushed samples were tested by NIR hyperspectral and processed using Multiplicative Scattering Correction (MSC) and Competitive Adaptive Reweighted Sampling (CARS). Kennard–Stone (KS) and Samples Set Partitioning Based on Joint X-Y Distance (SPXY) algorithms were used to select the optimal number of standard samples transferred by the model combined with root mean square error. The crushed samples were transferred into the complete samples prediction model through the Direct Standardization (DS) algorithm. The spectral hybrid model of crushed samples and full samples was established. The results showed that the Determination Coefficient of Calibration ($R_{P}^{2}$) value of the total samples prediction set increased from 0.5088 to 0.8068, and the value of the Root Mean Square Error of Prediction (RMSEP) decreased from 0.2454 to 0.1691 log_10_ CFU/g. After establishing the hybrid model, the RMSEP value decreased by 9.23% more than before, and the values of Relative Percent Deviation (RPD) and Reaction Error Relation (RER) increased by 12.12% and 10.09, respectively. The results of this study showed that TVC instewed beef samples can be non-destructively determined based on the DS model transfer method combined with the hybrid model strategy. This study provided a reference for solving the problem of poor accuracy and reliability of prediction models in heterogeneous samples.

## 1. Introduction

The total viable count (TVC) refers to the number of bacteria growing in samples per gram (per milliliter) under certain conditions (such as oxygen, nutritional conditions, pH, culture temperature, and time). The TVC test is used to determine the degree of bacterial contamination and food hygiene quality. And the result of the TVC test reflects whether the food meets the hygiene requirements and can make an appropriate hygiene evaluation of the tested samples [1,2]. To a certain extent, TVC indicates the quality of food hygiene [3,4]. In the daily diet of human beings, many bacteria can grow in the storage process after cooking. And a TVC exceeding the standard will greatly threaten human health [5]. Non-destructive determination of TVC in dishes by the spectroscopic method is an effective method for rapid evaluation of the freshness and edibility of dishes [6,7]. After cooking, the physical form of the samples will be deformed due to the change in the components of heat and mass transfer and other factors. The formed heterogeneous samples will have great differences in the spectrum’s reflection, scattering, and absorption, which seriously affects the accuracy and robustness of the established prediction model of TVC [5].

The freshness of meat dishes is related to the decomposition of nutrients such as water, protein, fat, and sugar. Under the joint action of enzymes and microorganisms [1], the hydrogen-containing groups in the tissue structure change. The spectral characteristic information reflects the absorption of these hydrogen-containing groups’ doubled and combined frequency. During the preservation process of meat, the protein, fat, and moisture content also change correspondingly, which affects the optical properties of meat tissue, such as spectral absorption coefficient and scattering coefficient [1]. Therefore, the spectral characteristics of organic compounds in meat could be used to rapidly detect meat quality characteristics and component content [8].

Near-infrared hyperspectral imaging (NIR-HSI) technology is a simple, rapid, and non-destructive analysis method. Its sample pretreatment is simple and environmentally friendly, consuming neither solvents nor producing chemical wastes. It has been widely used in tobacco [9,10], food [11], medicine [12], and other fields. For NIR-HIS analysis, the accuracy of the final results is closely related to the uniformity of samples. The accuracy and reproducibility of the final results would be better for some examples with uniform texture. Still, the accuracy and reproducibility of the analysis results would be poor for massive objects with uneven surfaces and composition distribution [5]. Research has shown that the uniformity of full samples (FS) is worse than that of crushed samples (CS) when measuring dishes using the diffuse mode [13,14]. There will be a significant difference in spectral scattering and absorption coefficient, and the accuracy and robustness of the TVC prediction model are consequently reduced. Therefore, NIR-HIS prediction models are primarily based on powder or mean samples. The samples are heterogeneous in many applications, such as online production, quality control, and in situ analysis. To ensure the accuracy of the model, model transfer (MT) and hybrid model (HM) strategies can be adopted to eliminate the influence of different sample states [15,16,17].

In the near-infrared spectrum field, MT strategies were widely used in model transfer between different types of spectrometers [18,19]. Over the past decade, many model transfer algorithms have been studied for model transfer between different spectrometers. Typical model transfer algorithms include Direct Standardization (DS) [20], Piecewise Direct Standardization (PDS), Double Window Piecewise Direct Standardization (DWPDS) [21], and Orthogonal Signal Correction (OSC) [22]. These algorithms could also be applied to transfer models between different physical states of the same substance [23,24,25]. Pereira et al. [23] applied the DWPDS method to model the delivery of drug powder mixtures to complete tablet samples. They successfully applied the Partial Least Squares (PLS) model based on powder mixtures to complete tablet prediction of nevirapine. Liu et al. [24] proposed a new method for predicting protein and amylose contents in rice. They used the spectrum transfer based on DS, transforming the spectrum of particles into the corresponding powders, and successfully achieved the prediction of grain rice with the powder model. Munnaf et al. [25] took soil texture as the research object and explored whether a short-range near-infrared reflectance spectroscopy (Vis-NIRS) sensor can classify soil texture accurately based on the PDS and DS. The accuracy of soil texture classification using a short-ranged Vis-NIRS sensor was improved successfully. The content of nicotine in marine sediment carbon content was predicted. The above research was mainly applied to the transfer between sample models in regular physical form. There were few studies on the model transfer of meat dishes, a complex physical form. In the previous study [5], our team successfully detected the TVC in the crushed samples of stewed beef and found that the modeling effect of the complete samples was lower than that of the crushed samples. We continued the team’s previous work to solve the problem of poor accuracy and reproducibility of the complete sample prediction model. We discussed the feasibility of using MT and HM to improve the prediction performance of the total number of bacteria in intact stewed beef samples.

In this study, the research object was stewed beef dishes with different storage times, and the method of establishing HM based on CS and FS after the transfer was studied to achieve non-destructive determination of TVC in heterogeneous samples of meat dishes. It is mainly divided into the following contents:Comparison of CS and FS prediction model performance based on NIR-HIS and SVM algorithm.Selection of optimal standard sample number.Taking advantage of the DS algorithm to perform MT between CS and FS prediction models.Performing hybrid modeling of the BS model and the posttransfer model to finish the prediction of the total number of bacteria in stewed beef samples.

## 2. Materials and Methods

### 2.1. Experimental Materials and Reagents

#### 2.1.1. Preparation of Experimental Materials

Fresh bovine upper brain and seasoning were purchased from Xingfurongyao Supermarket in Haidian District, Beijing. The samples of 1.8912 kg fresh bovine upper brain were transferred to laboratory shortly after being vacuum packaged and stored in an ice bath. The seasoning was 101 mL oil, 20 g salt, and 50 mL soy sauce. The preparation process of the complete samples was as follows: the upper brain of cattle was cut into cubes of dimensions 2.5 cm × 2.5 cm × 2.5 cm, and the quantity was 100 pieces. The upper brains of cattle were boiled in cold water for 10 min to remove blood foam and stir-fried in a pot with hot oil. Then, 1500 mL water was added. Subsequently, the seasonings were added, and the mixture was braised for 1.5 h. Finally, the high fire was used to boil the water. The finished samples were placed in a tray and covered with cling film. Then, these samples were stored in the refrigerator at 4 °C. Ten stewed beef samples were taken out every 24 h to collect NIR-HIS data of the complete morphology. Then, the complete sample was crushed by the cooking machine (2500 rpm for 15 s) and placed in Φ 35 mm Petri dishes. NIR-HSI data of samples in the crushed form were measured again after compaction. Finally, the total colony content of the samples was determined.

#### 2.1.2. The Main Reagent of the Experiment

Sterile saline was obtained from Shijiazhuang No. 4 Pharmaceutical Co., Ltd. (Shijiazhuang, China). A 3 M^TM^ Petrifilm^TM^ aerobic count plate was purchased from Minnesota Mining and Manufacturing Co., Ltd. (Saint Paul, MN, USA).

### 2.2. Experimental Method

#### 2.2.1. Determination of Total Bacterial Colony

The method used in this study was plate colony counting [26], the detection method was operated according to China National Standard GB/T 4789.2 (2022), and the specific operation was to dilute the samples to be tested at the ratios of 1:10 and 1:100. The microbes were fully dispersed into individual cells. A certain amount of dilution solution was coated on the test tablet. After being cultured, each cell grew and multiplied to form a colony visible to the naked eye. A single colony should represent a single cell in the original samples. Finally, the number of bacteria in the sample was calculated based on the dilution rate of the sample and the sample inoculum size. The calculation formula was as follows:(1)$$N=\frac{\sum C}{\left(n_{1}-0.1n_{2}\right)d}$$
where N represents the number of colonies in the sample, C represents the number of colonies on the plate, $n_{1}$ represents the number of plates at 10 times dilution, $n_{2}$ represents the number of plates at 100 times dilution, and d represents the dilution factor. Finally, the number of bacteria in the sample was log-transformed and expressed in log_10_ CFU/g for further analysis.

#### 2.2.2. Near-Infrared Hyperspectral Image System

The schematic diagram of the NIR-HIS imaging system was shown in Figure 1, which consists of an imaging spectrometer (ImspectorNl7E spectrometer with PDA detector), illuminating system (10 W halogen lamp), conveying unit (samples tray and linear conveying platform), computer control unit (i7 processor, 16 GB of memory), and other attachments (OLE15lens, focusing plate and Teflon White reference plate). The spectrometer collected spectral images in the wavelength range of 885–1735 nm with a spectral resolution of 3.3 nm. After scanning each sample, we obtained NIR-HIS images with 256 wavelengths.

#### 2.2.3. Hyperspectral Image Acquisition and Spectra Extraction

The NIR-HIS collection of stewed beef samples was conducted indoors daily from 10:00 to 12:00 (Beijing time). The spectra of full and crushed samples were collected sequentially. The hyperspectral imaging system was switched on and preheated for 30 min. The exposure time was 8 s, with a scanning rate of 32 mm/s and a scanning area of 50%. The reference whiteboards (100% reflectivity) for white balance were collected, then the lens was turned off, and the images (0% reflectivity) for black balance were capture d. Formula (1) was used to obtain the NIR-HSI reflectance after correction of the original spectral data. During collection, all samples were placed on the conveyor belt at the same time each day, and the full and crushed samples were collected, respectively. Each batch of samples was collected 3 times repeatedly, and the average value was taken as the signal value. Prediktera Envince 2.7.11 software was used for hyperspectral data collection. As shown in Figure 2, the center area within 2 mm of the edge of each sample was selected as the region of interest (ROI), and the mean spectral reflectance of all pixels in the ROI of the samples scanned three times was used as the original data of each spectrum.
(2)$$I_{norm}=\left(\frac{I_{raw}-I_{dark}}{I_{white}-I_{dark}}\right)\times 100\%$$

In Formula (2), $I_{raw}$ stands for the original sample image, $I_{white}$ stands for the white balance image, $I_{dark}$ stands for the black balance image, and $I_{norm}$ stands for the corrected image. 

### 2.3. Date Processing

#### 2.3.1. Preprocessing Method of Original Spectral Data and Model Establishment

The original spectra were pretreated by the MSC method to reduce the effect of interference, such as instrument baseline shifts, sample surface inhomogeneities, and other factors [27]. To simplify the model and heighten the prediction accuracy of the model, Competitive Adaptive Reweighted Sampling (CARS) was used to screen the characteristic wavelengths [28,29]. The prediction model was established by using SVM [30]. ɛ-SVR in the Lisvm toolbox was used in this study. The ɛ-SVR can achieve regression with strong robustness by introducing ɛ-insensitive loss function.

#### 2.3.2. Model Performance Evaluation

Determination Coefficient of Calibration ($R_{C}^{2}$), Cross-Validation ($R_{CV}^{2}$), Prediction ($R_{P}^{2}$), Root Mean Squared Error of Calibration (RESEC), Root Mean Square Error of Cross-Validation (RMSECV), Root Mean Squared Error of Prediction (RMSEP), Relative Percent Deviation (RPD), and Ratio Error Range (RER) were used to evaluate the performance, accuracy, and stability of the model [31]. In addition, Latent Variables (LVs) is also an important parameter that affects model performance. When R^2^ approaches 1, RMSEC and RMSEP become smaller and closer, indicating that the calibration set’s spectral information is fully extracted and the model has good stability, high fitting degree, and good prediction ability [32,33]. RPD could measure the reliability of the quantitative correction model of NIR spectroscopy. It is generally considered that the model is more reliable when the RPD > 2.5 [34], a model with a RER below 3 has little practical utility, and a model with a RER between 3 and 10 has good practical utility. Those above 10 have high efficiency [35]. The calculation method for each indicator was as follows:(3)$$RMSE=\sqrt{\frac{\sum_{i=1}^{n}{\left(y_{i}-{\hat{y}}_{i}\right)}^{2}}{n}}$$

In Formula (3), y_i_ and ${\hat{y}}_{i}$ are the measured value and predicted value of indicators, respectively.
(4)$$RPD=\frac{SD}{RMSEP}$$

Here, SD stands for the standard deviation of the analysis samples, and RMSEP stands for the verified root mean square error of the analysis samples.
(5)$$RER=\frac{Max-Min}{RMSEP}\;$$

In addition, Max and Min are the maximum and minimum values of the prediction set, respectively.

Although the physical morphology of the two groups of samples was different and the selected characteristic wavelength was not completely the same, the two groups belong to the same samples and the selected characteristic wavelength was closely related to the total colony content of the measurement index. In this study, feature wavelengths screened in different physical forms were combined into a new variable set, and only one of the same feature wavelength sets was reserved for subsequent analysis.

#### 2.3.3. Model Transfer

MT means that after mathematical processing, the model of host or source samples could be used in the target machine or target samples to achieve model sharing and effective utilization [36,37]. This study used the DS algorithm to conduct MT on stewed beef samples with two physical forms [24,37]. The DS algorithm can perform direct correction of spectral data. While the host or source sample model remained unchanged, the spectrum of the target machine or target samples was corrected using the transformation matrix F to match the spectrum of the host or source samples. The solving process of transfer matrix F was as follows:(6)$${MS}_{m\times n}={SS}_{m\times n}\times F_{n\times n}+E_{m\times n}$$
(7)$$E_{m\times n}=O_{m\times 1}\times {b_{n\times 1}}^{T}$$
where ${MS}_{m\times n}$ is the transfer samples spectral matrix measured by the host, ${SS}_{m\times n}$ is the transfer samples spectral matrix measured by the target machine, $E_{m\times n}$ is wavelength residual matrix, each line is equal, $O_{m\times 1}$ is a column vector with all 1’s, $b_{n\times 1}$ is the calibration matrix for each wavelength point, and T is an n-dimensional column vector.
(8)$$C_{m\times m}=I_{m\times m}-\left(\frac{1}{m}\right)\times O_{m\times 1}\times {O_{m\times 1}}^{T}$$
where $C_{m\times m}$ is the centralized matrix, and $I_{m\times m}$ is the identity matrix. Both sides of Equation (6) are multiplied by Equation (8). Equation (6) can be simplified as follows:(9)$$S_{ms}=S_{ss}\times F$$
where $S_{ms}$ is the centralized matrix of the matrix ${MS}_{m\times n}$, and $S_{ss}$ is the centralized matrix of the matrix ${SS}_{m\times n}$. The following formula can solve the transformation matrix F:(10)$$F=S_{ss}^{+}\times S_{ms}$$
where $S_{ss}^{+}$ is the generalized inverse matrix of the matrix $S_{ss}$. There were many wavelengths in NIR spectral data, and the number of samples used for MT was generally much smaller than the number of wavelengths. Therefore, there would be ill-conditioned problems when directly obtaining the generalized inverse matrix of $S_{ss}$ which would affect the MT effect. Singular value decomposition (SVD) is often used in practical applications; for details, please refer to reference [20].

After obtaining the value of $S_{ss}^{+}$, the transformation matrix F can be obtained according to Formula (10). To predict the unknown samples $X_{un}$, the transformation matrix F is used to obtain the converted spectrum $X_{t}$, as shown in Formula (11):(11)$$X_{t}=X_{un}\times F$$

Then, the model is established by the host computer to predict $X_{t}$.

#### 2.3.4. Selection of the Transfer Samples

The DS algorithm belongs to the MT algorithm with standard samples, and it is necessary to select standard samples from the training set to form the standard sample set to calculate the transformation matrix. The quality and quantity of the standard sample set directly affect the effect of spectral transmission. To ensure that the selected sample set was representative, the Kennard–Stone algorithm (KS) and Samples Set Partitioning Based on Joint X-Y Distance algorithm (SPXY) were used to determine standard 0 samples [38]. The KS algorithm considers all samples as candidate samples and selects samples from them in turn to enter the standard samples set: first, the two samples with the longest Euclidean distance are selected into the standard samples set, then the Euclidean distance between each remaining sample and each known sample in the standard samples set is calculated, the candidate samples with the maximum and minimum distance are found and put into the standard samples set, and so on, until the desired number of samples is reached. The SPXY algorithm is developed based on the KS algorithm, which considers both the X variable (spectral variable) and the Y variable (physicochemical value variable) when calculating the distance between samples (12)–(14).
(12)$$d_{x}\left(p,q\right)=\sqrt{\sum_{j=1}^{N}{\left[X_{p}\left(j\right)-X_{q}\left(j\right)\right]}^{2}}p,q\in \left[1,Z\right]$$
(13)$$d_{y}\left(p,q\right)=\sqrt{{\left(y_{q}-y_{p}\right)}^{2}}=\left|y_{p}-y_{q}\right|p,q\in \left[1,Z\right]$$
(14)$$d_{xy}\left(p,q\right)=\frac{d_{x}\left(p,q\right)}{\underset{p,q\in \left[1,N\right]}{\max}d_{x}\left(p,q\right)}+\frac{d_{y}\left(p,q\right)}{\underset{p,q\in \left[1,N\right]}{\max}d_{y}\left(p,q\right)}p,q\in \left[1,Z\right]$$

In Formulas (13) and (14), $X_{p}$ and $X_{q}$ represent the spectral variables of two different samples, $y_{p}$ and $y_{q}$ represent the physicochemical variables of two different samples, and Z represents the number of spectral wave points of the samples.

Some studies have pointed out that the transmission results of different indicators may be related to the standard sets selected from various forms during transmission [24]. In this study, an attempt was made to select standard samples from the calibration set of broken and intact physical samples, select the different quantities of standard samples to form the standard sample set, and compare the MT effect of two standard sample sets. It should be pointed out that the corresponding samples must be selected from the complete samples when a standard sample is selected from the calibration set of crushed samples. The corresponding samples must be selected from the crushed samples when a standard sample is selected from the complete sample calibration set.

#### 2.3.5. The Establishment of a Hybrid Model

A hybrid model (MM) is a model that expends the spectral principal component space by adding variation samples to the basic model, making it more accurate and robust, with better predictive capabilities and broader forecast horizons [39]. In this paper, a mixture model of crushed and intact stewed beef samples was established, and the principal component space of the spectrum of the crushed samples model was expanded by using the range of the complete stewed beef samples. The original CS model was remodeled by adding the FS spectrum before and after the transfer to more accurately predict the TVC value of the complete stewed beef.

## 3. Results

### 3.1. Results of Microbiological Analyses

Figure 3 depicts the reference the TVC values of beef dishes tested on each day of the experiment. During the storage period, the TVC value showed an upward trend and had a certain significance (*p* < 0.05). The TVC value reached the maximum value of 3.79 log_10_ CFU/g on the 10th day.

Before establishing the prediction model, the 100 samples were randomly divided into a calibration and prediction set in the ratio of 4:1. Statistics of the measurement results of beef dishes’ TVC values in the calibration set and prediction set are shown in Table 1. No significant difference was found between the calibration and prediction set’s average and standard deviation. Thus, the division of the samples was reasonable.

### 3.2. Difference Analysis of Spectral and Modeling Results

The original spectral reflection curves of CS and FS are shown in Figure 4. It can be seen from Figure 4a that the spectral peak shapes of different crushed samples were roughly the same, and the overall trend was similar. There were no obvious abnormal samples. The bands had many obvious peaks and valleys near 944 nm, 980 nm, 1076 nm, 1186 nm, 1264 nm, and 1455 nm. These absorption bands were formed by the overtone and combination vibrations of the molecular chemical bonds in the samples. The spectral data contained a lot of information about the content of the different components [40]. The N-H stretching vibration had a second overtone near 1076 nm. A third overtone absorption band of the C-H vibration near 1186 nm could obtain protein-related information. At 1264 nm was the stretching vibration absorption band of the C-H group, which was related to fat content. The water molecules may have produced a peak at approximately 950 nm, which is associated with a second overtone frequency doubling absorption band of the O-H group. Another broad peak was located at approximately 1455 nm, possibly associated with the first overtone of the O-H stretching [41]. It can be seen from Figure 4b that the peak shape of FS was consistent overall, with a few anomalies which may be related to the uneven surface tissue of the samples. The peak shape was less evident than that of the crushed samples, and smaller troughs or peaks could still be observed in the above bands. In the early stage, because the beef was rich in nutrients, bacteria multiplied rapidly, and the total number of colonies increased rapidly. As bacteria grow in beef, many of the nutrients in the meat are crushed down. With the oxidation of nutrients such as protein and fat and the accumulation of metabolites, bacterial growth slows or even declines. Therefore, the change in beef internal substances (protein, fat, etc.) was related to the change in the total number of bacteria.

The spectra pretreated by MSC can eliminate the effects of light scattering caused by uneven particle size and linear translation in the spectrum, enhance the spectrum specificity, and improve the modeling effect significantly [27]. The relationship between diffuse absorbance and sample component content can be simulated as a linear relationship when the scattering coefficient of samples remains unchanged, and the samples’ component content is within a specific range. Changes in the physical state of the samples would lead to changes in the intercept and slope of the linear relationship, resulting in changes in a linear relationship between the diffuse absorbance and the ideal absorbance (average absorbance). The MSC method was based on this idea, using the average spectrum as the standard spectrum, performing linear regression between the spectral band of each sample and the standard spectrum, and then correcting each sample through reverse calculation [28]. It can be seen from Figure 4c,d that the offset can be improved due to light scattering after MSC pretreatment. However, the difference was still obvious for the band of FS, especially in the spectral range of 900 nm–1150 nm. Therefore, there were certain deviations and changes in the spectra of samples with different physical forms, which pretreatment could not eliminate.

Figure 5 shows the spectrum’s Principal Component Analysis (PCA) score under the two physical forms. Solid blue squares represented crushed samples, and hollow dots represented full samples. From the figure, it can be seen that the spectra of both samples have undergone certain changes during storage. The spectral difference between the two physical forms was pronounced during storage, indicating that the change in the material form of the samples would lead to a change in the spectrum. Spectral modeling of samples with different physical forms may lead to an unusable model or a large deviation in prediction results.

The MSC method was used to preprocess the BS spectra, and the CARS algorithm was used to screen feature variables. The results show that the established SVM prediction model has a good prediction effect. Table 2 illustrates the comparison results of TVC content prediction models established by using the same pretreatment and characteristic variable selection method for the two physical morphology spectra. As the table shows, the CS model’s prediction performance was better than that of the FS. Therefore, MT technology was considered to correct spectra of different physical forms, which can improve FS prediction performance.

### 3.3. Model Transfer Analysis

As shown in Figure 6, we summarized the process of detecting the total number of bacterial colonies in beef dishes using a hyperspectral mixture model. In the first step, this study used MSC and CARS to realize spectral data preprocessing and characteristic wavelength selection. In the second step, we used KS and SPXY algorithms combined with root mean square error to select the optimal number of standard samples transferred by the model. The principle of the KS algorithm is to regard all samples as candidate samples of the calibration set and select samples from them in order to enter the calibration set. It has the outstanding advantages of being fast, efficient, simple, intuitive, and highly representative when selecting samples. It can also ensure that the samples in the calibration set are evenly distributed according to spatial distance. The SPXY algorithm is developed based on the KS algorithm. It can consider both x and y variables when calculating the distance between samples, and it can increase the difference and representativeness between samples, thereby reducing the number of samples in the calibration set and the amount of calculations in the modeling process. Finally, the DS algorithm was used to transfer the crushed samples to the complete sample prediction model. And the spectral hybrid model was established for the crushed samples and full samples, which improved the stability of the model.

#### 3.3.1. The New Set of Variables and the Predicted Results for Two Physical Forms

Two groups of variables were selected to form a new set of variables for screening key variables. The reason was that it was different for the variables screened by the two forms. Table 3 shows that the closely related characteristic variables were screened out by CARS.

Figure 7 shows the screening process of CARS variables for two physical morphology samples. As the sampling time increases, the number of selected variables gradually decreases. Meanwhile, the RMSECV values of 10-fold cross-validation firstly decreased to the lowest point and then increased rapidly. The independent wavelength variable of the TVC was excluded during the process, and the RMSECV value decreased. And then the wavelength variables related to the composition were removed, and information loss caused the RMSECV value to increase. When the CS model was run 24 times, the RMSECV value was the minimum, and 19 characteristic wavelength variables were screened related to TVC content. When the FS model was run 37 times, the RMSECV value was minimum, and 7 characteristic wavelength variables were screened related to TVC content.

Table 4 shows the two samples’ modeling results using the variable set’s spectral bands. It can be seen that compared with the previous modeling results, the $R_{P}^{2}$ of the FS prediction set increased from 0.3453 to 0.5088, but the results of the RMSE and RPD were not ideal. The RPD was still less than 2.5, and the reliability of the model was poor.

#### 3.3.2. Selection of Optimal Standard Sample Number and Establishment of Transfer Model

DS is a standard algorithm, and the selection of standard samples would affect the final prediction effect. Therefore, the standard sample numbers (SSN) should be optimized before MT analysis. The optimal SSN was determined by the principle of minimum RMSEP in the application process. Due to the difference between the spectrum of the crushed samples and the full samples, there is a difference between the transfer of standard samples obtained by using two groups of samples as standard sets and choosing the same number of SNN, and the transfer effects of spectra would be different. In addition, the results of this study indicate that the transfer results of different indexes may be related to the difference in standard sets selected from different physical forms during transfer. This result was consistent with the research results of Liu et al.’s study [38]. In this study, standard samples were selected from two schemes. Namely, the crushed correction samples were used as the standard set, the full correction samples were used as the dependent set, and the crushed correction samples were used as the dependent set. The full correction samples were used as the standard set, and then the optimal scheme was selected for subsequent MT.

Figure 8 shows the influence of different SSN on the RMSEP value of the prediction results of the algorithms with the crushed correction sample as the standard set. It can be seen from the figure that the RESEP values of the two algorithms fluctuated continuously within 35 SSN, fluctuating up and down, and then decreasing rapidly until reaching the lowest point at 70 SSN. Based on comprehensive consideration, it was better to choose 70 standard samples.

Table 5 shows the modeling results of selecting the optimal SSN for the FS based on the crushed correction samples as the standard set. Compared with the results in Table 3, the prediction effect of the model has been further improved. Among them, the results after the SPXY algorithm selected the standard sample have been obviously improved. According to the results of selecting the optimal SSN from the standard set of crushed correction samples, the value of $R_{P}^{2}$ was 0.8069, the value of RMSEP was 0.1691 log_10_ CFU/g, the value of RPD was 2.98, and the value of RER was 10.01. The standard samples selected by the SPXY algorithm had a better modeling effect on FS.

Figure 9 presents different SSNs’ influence on the predicted results’ RMSEP value. As can be seen from the figure, the RMSEP value of the two algorithms showed a decreasing trend in the oscillation. The RMSEP value of the model was the lowest, which was established using 60 standard samples selected by the KS algorithm and 80 standard samples selected by the SPXY algorithm.

Table 6 reveals the modeling results of the optimal SSN for the FS selected with the full calibration sample as the standard set. Also, the modeling results of the FS from the two algorithms show that all of the indexes were significantly improved, the SPXY algorithm was still the better one, and it was not obvious that there was a difference between the two algorithms. There was a big gap in the prediction accuracy compared with the best standard samples selected from the standard set.

According to the results given in Table 5 and Table 6, the SPXY algorithm is a feasible and effective method to select standard set samples from crushed calibration samples and transfer the TVC model.

Figure 10 shows the PCA score of the spectrum for the latter two physical forms of MT. The blue square represents the crushed sample and the hollow origin represents the complete sample. After MT, the spectral difference between the two physical forms has significantly improved. It shows that the MT method can overcome the spectral changes caused by the physical morphology changes in the samples to a certain extent and can narrow the differences between them. At the same time, the FS’ spectrum principal component differentiation was more obvious after transfer, which may be due to the better predictive performance of the FS model after transfer.

#### 3.3.3. The Establishment of the Hybrid Model

In the last section, the transfer model with the CS as the host and the FS as the target was established by spectral pretreatment and the MT method, which reduced the error of TVC directly predicted by the CS model and improved the prediction accuracy. In order to further improve the prediction accuracy of the model and reduce the prediction error, this experiment took the crushed samples as the basic model and added a certain number of the FS, respectively, to expand the scope of principal component space covered by model calibration set samples. MM was established to continue to improve the ability of the crushing model to predict the FS. Table 7 and Table 8 show the prediction results of the FS by directly adding the FS’ spectrum and by adding the FS’ spectrum after transfer, respectively. It can be seen from Table 7 that the best prediction results were obtained by adding 20 FS’ spectra to the CS’ spectra directly, which expanded the coverage of the calibration set sample. The prediction ability has been improved to some extent compared with that without the addition, but the overall prediction ability was limited, which is still inferior to the result after MT in Table 6.

It can be seen from Table 8 that the best prediction results were obtained by adding 15 transferred FS’ spectra to the CS’ spectra. The $R_{p}^{2}$ value was similar to using MT only. The RMSEP decreased by 6.68% and 9.23%. The RPD increased by 4.55% and 12.12%, respectively, compared with the value before HM establishment. The stability and reliability of the model were enhanced. The RER was greater than 10, indicating that the model has high efficiency. Therefore, adding a certain amount of the FS’ spectrum in the calibration set can improve the prediction ability of the HM for complete stewed beef samples. The effectiveness of the HM was certified, and a good prediction effect was achieved. In conclusion, the stability and reliability of the model can be improved by adding a certain amount of transferred full samples to the crushed sample model as the modeling set, thereby improving the prediction accuracy of TVC in the complete stewed beef samples.

## 4. Conclusions

In this study, in order to solve the problem of poor accuracy, stability, and reliability of the prediction model when measuring massive FS of stewed beef dishes by NIR-HIS in diffuse reflection mode, a solution based on CS combined with the MT method was proposed, and a hybrid model was established. The SPXY algorithm was used to take the crushed correction sample as the standard sample, and the standard samples of MT were selected from the calibration set samples of crushed and full stewed beef samples. When 70 standard samples were selected, $R_{P}^{2}$ = 0.8069, RMSEP = 0.1691 log_10_ CFU/g, RPD = 2.98, and RER = 10.01. The DS algorithm was used to calculate the transformation matrix to achieve the spectral transfer between different physical forms of samples. Compared with the FS modeling effect using the SVM model directly, the prediction effect of the FS after MT is significantly improved. Compared with before model mixing, the RMSEP value of the HM of cooked beef samples decreased by 9.23%, and the value of RPD increased by 12.12%. To further improve the prediction accuracy of the model and reduce the prediction error, two kinds of physical HMs were established. The training set samples were based on the CS’ and FS’ spectrum added before and after incremental transfer to establish the HM of two physical forms. The results show that the stability and reliability of the HM based on the CS’ and FS’ spectrum after transfer were improved. In practical application, the model established by the above method can simplify the sample pretreatment process, save time and workload, and achieve non-destructive accurate and rapid detection of the TVC of the massive dishes sample. 

## Figures and Tables

**Figure 1 sensors-23-09584-f001:**
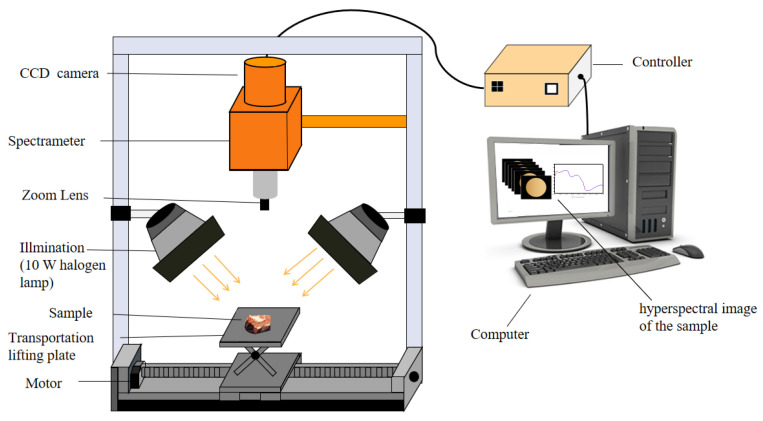
Schematic of the main components of the NIR-HIS imaging system.

**Figure 2 sensors-23-09584-f002:**
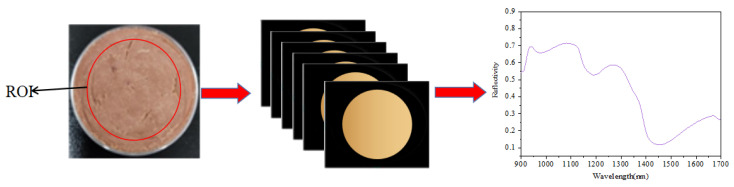
Hyperspectral images of ROI.

**Figure 3 sensors-23-09584-f003:**
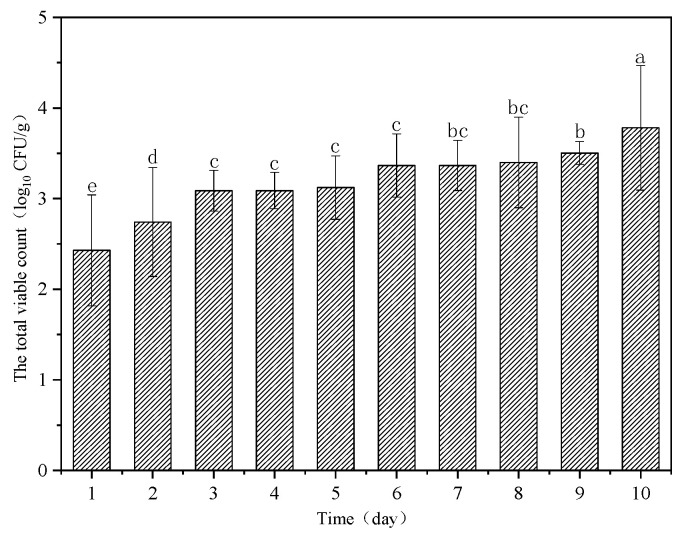
Reference beef dishes’ TVC values tested on each day.

**Figure 4 sensors-23-09584-f004:**
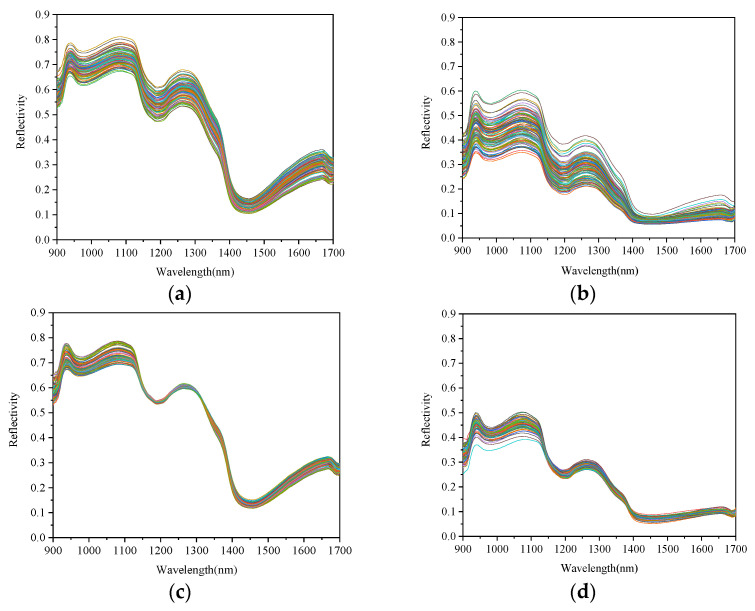
The original spectrum of CS and FS (**a**,**b**) and spectra of CS and FS after MSC pretreatment (**c**,**d**).

**Figure 5 sensors-23-09584-f005:**
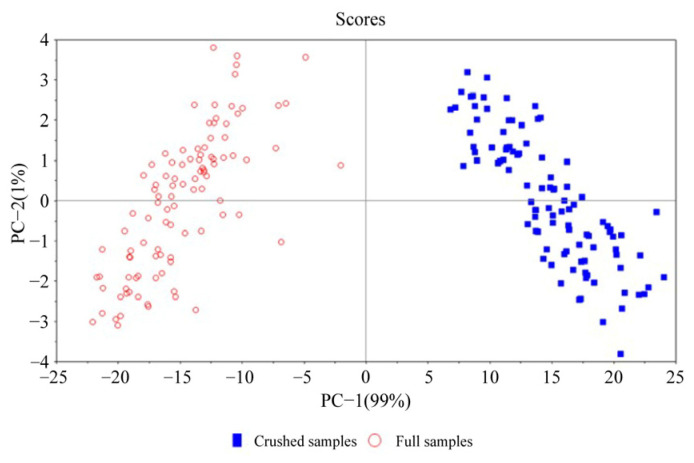
Principal component analysis scores of CS and FS spectra.

**Figure 6 sensors-23-09584-f006:**
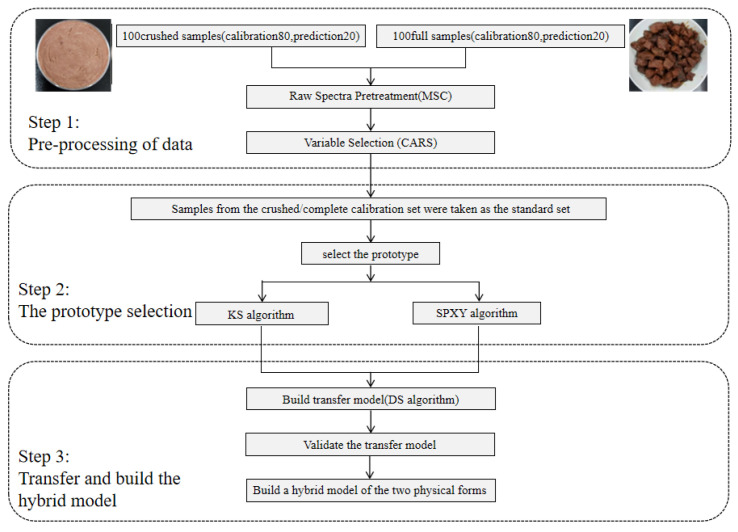
Flow chart for prediction of total bacterial count in stewed beef by using hyperspectral mixture model.

**Figure 7 sensors-23-09584-f007:**
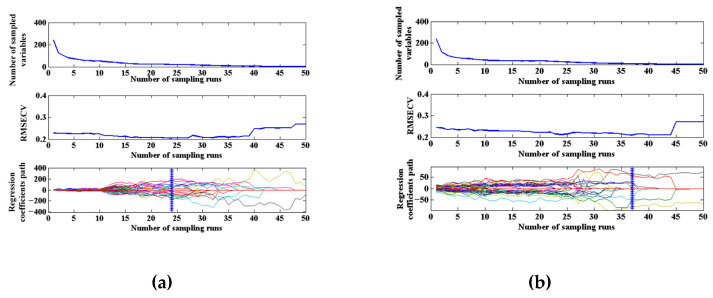
The screening process of CARS variables for two physical morphology samples: (**a**) CS; (**b**) FS.

**Figure 8 sensors-23-09584-f008:**
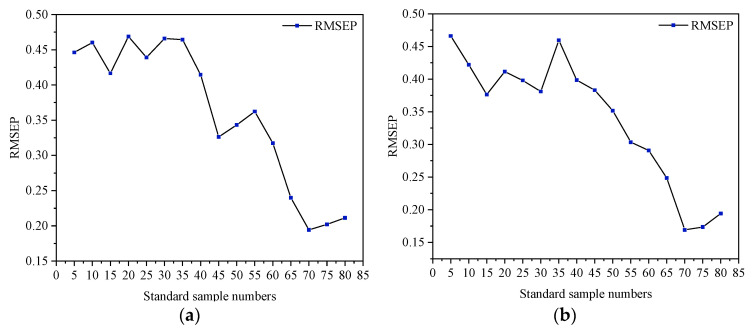
Selection of MT optimal SSN based on crushed calibration samples as the standard set. (**a**) KS algorithm; (**b**) SPXY algorithm.

**Figure 9 sensors-23-09584-f009:**
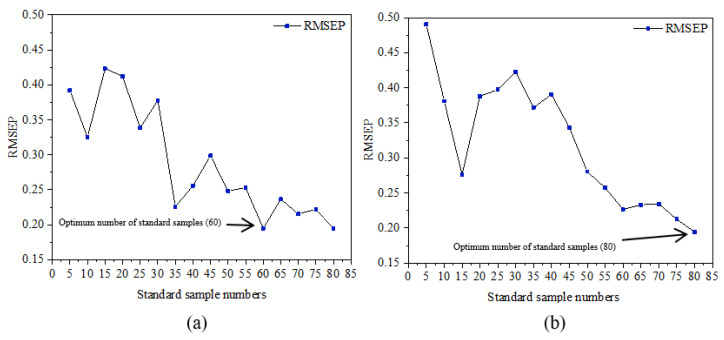
Selection of MT optimal SSN based on complete calibration sample. (**a**) KS algorithm; (**b**) SPXY algorithm.

**Figure 10 sensors-23-09584-f010:**
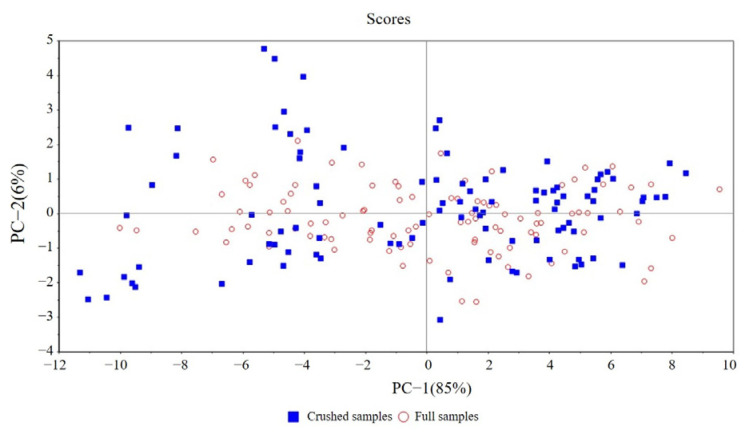
Difference diagram of principal component scores of CS and FS in NIR hyperspectral model after MT.

**Table 1 sensors-23-09584-t001:** Measurement results of beef dishes’ TVC values in the calibration and prediction sets.

Samples Type	Samples Size	The Maximum/ (log_10_ CFU/g)	The Minimum/ (log_10_ CFU/g)	The Average/ (log_10_ CFU/g)	Standard Deviation/ (log_10_ CFU/g)
Calibration set	80	3.8808	2.1347	3.1140	0.2953
Prediction set	20	3.7559	2.1047	3.1490	0.3623

**Table 2 sensors-23-09584-t002:** Calibration and prediction results of the TVC values of beef dishes for two physical morphology forms by using NIR hyperspectral model.

Samples	Pretreatment Method	Variable Filtering Method	LVs	Calibration Set		Cross-Validation		Prediction Set			
				$R_{c}^{2}$	RMSEC	$R_{cv}^{2}$	RMSECV	$R_{p}^{2}$	RESEP	RPD	RER
CS	MSC	CARS	10	0.6603	0.1887	0.6650	0.1954	0.6438	0.2037	2.78	8.11
FS	MSC	CARS	12	0.6128	0.2005	0.5091	0.2385	0.3453	0.2713	2.34	6.08

**Table 3 sensors-23-09584-t003:** Characteristic variables of two physical forms were screened by the CARS algorithm.

Samples	Characteristic Variables (nm)
CS	954, 977, 994, 1000, 1027, 1030, 1040, 1057, 1067, 1117, 1120, 1213, 1237, 1243, 1247, 1260, 1310, 1327, 1685
FS	1037, 1080, 1140, 1183, 1648, 1681, 1691

**Table 4 sensors-23-09584-t004:** Calibration and prediction results of the TVC values of beef dishes for two physical morphology forms by using NIR hyperspectral model under the new set of variables.

Samples	Pretreatment Method	Variable Filtering Method	LVs	Calibration Set		Cross-Validation		Prediction Set			
				$R_{c}^{2}$	RMSEC	$R_{cv}^{2}$	RMSECV	$R_{P}^{2}$	RMSEP	RPD	RER
CS	MSC	CARS	8	0.8424	0.1258	0.7946	0.1653	0.6514	0.2032	2.67	8.33
FS	MSC	CARS	11	0.5336	0.2093	0.5217	0.2281	0.5088	0.2454	2.37	6.90

**Table 5 sensors-23-09584-t005:** The modeling results of optimal SSN for FS were selected from the standard set of crushed correction samples.

Method	SSN	$R_{c}^{2}$	RMSEC	$R_{p}^{2}$	RMSEP	RPD	RER
KS	70	0.8424	0.1258	0.6312	0.1962	2.71	8.63
SPXY	70	0.8424	0.1258	0.8069	0.1691	2.98	10.01

**Table 6 sensors-23-09584-t006:** The modeling results of the optimal SSN for FS were selected with the full calibration sample as the standard set.

Methods	SSN	$R_{c}^{2}$	RMSEC	$R_{p}^{2}$	RMSEP	RPD	RER
KS	60	0.8424	0.1258	0.6519	0.1946	2.72	8.70
SPXY	80	0.8424	0.1258	0.6553	0.1942	2.72	8.72

**Table 7 sensors-23-09584-t007:** CS and FS hybrid model predict the results of FS.

Based	Added	$R_{c}^{2}$	RMSEC	$R_{p}^{2}$	RMSEP	RPD	RER
Spectra of 80 crushed samples	5	0.8802	0.1164	0.3958	0.5985	0.55	2.83
	10	0.9364	0.0909	0.0057	0.3591	0.96	4.71
	15	0.5740	0.2308	0.4593	0.2571	2.36	6.58
	20	0.5889	0.2256	0.5564	0.2181	2.59	7.75
	25	0.9036	0.1089	0.4446	0.2991	2.16	5.65
	30	0.8836	0.1200	0.4347	0.2598	2.31	6.51
	35	0.9174	0.0986	0.2521	0.2882	2.67	5.87
	40	0.8140	0.1483	0.2271	0.2873	2.15	5.89
	45	0.9095	0.1024	0.1495	0.3252	2.02	5.20
	50	0.8953	0.1077	0.0363	0.3728	0.87	4.54
	55	0.8332	0.1371	0.1349	0.3422	0.94	4.94
	60	0.8174	0.1405	0.0931	0.3461	0.91	4.89
	65	0.8234	0.1385	0.0671	0.3541	0.89	4.78
	70	0.5479	0.2103	0.6285	0.2145	2.45	7.88
	75	0.4640	0.2247	0.7408	0.1877	2.63	9.01
	80	0.5401	0.7349	0.6527	0.2109	2.45	8.02

**Table 8 sensors-23-09584-t008:** The prediction results of FS using CS and transferred FS hybrid model.

Based	Added	$R_{c}^{2}$	RMSEC	$R_{p}^{2}$	RMSEP	RPD	RER
Spectra of 80 crushed samples	5	0.8319	0.1395	0.7926	0.1528	2.07	10.72
	10	0.8192	0.1476	0.7355	0.1841	2.79	9.19
	15	0.8010	0.1573	0.8019	0.1535	2.22	12.02
	20	0.7590	0.1705	0.7411	0.1725	2.96	9.80
	25	0.7877	0.1569	0.7360	0.1886	2.76	8.97
	30	0.8836	0.1200	0.4347	0.2598	1.31	6.51
	35	0.8301	0.1450	0.6079	0.2278	2.52	7.42
	40	0.8210	0.1459	0.6030	0.2210	2.54	7.65
	45	0.8008	0.1531	0.6266	0.2207	2.54	7.66
	50	0.8028	0.1499	0.6406	0.2183	2.53	7.75
	55	0.7468	0.1687	0.7702	0.1786	2.84	9.47
	60	0.7269	0.1714	0.7405	0.1900	2.70	8.90
	65	0.7732	0.1529	0.8174	0.1766	2.80	9.58
	70	0.7293	0.1652	0.6954	0.2066	2.52	8.19
	75	0.7300	0.1626	0.7467	0.1960	2.58	8.63
	80	0.7403	0.1594	0.8150	0.1692	2.81	9.99

## Data Availability

Data will be made available on request.

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
