# Peer review of "Rapid Detection of Total Viable Count in Intact Beef Dishes Based on NIR Hyperspectral Hybrid Model"

_sensors, 2023, doi:10.3390/s23239584_

Round 1
Reviewer 1 Report
Comments and Suggestions for Authors
The author's research is meaningful for ensuring the quality of beef. However, there are some issues in the calibration process. Firstly, establish a model using formal chemometrics, and then evaluate the model using calibrated statistical parameters. Finally, use external data to validate the model.
Comments:
1. Model transfer is meaningless for this study, please eliminate it.
2. For the calibration, the main factors, R2cv, and RMSECV should be given, and they are very important parameters for the evaluation of models.
3. There are many over-fittings for the calibration, e. g. Table 4 (CS), Table 5, Table 6, RMSEPs were much larger than RMSECs.
4. Table 1, for the unit (1g CFU/g), “1g” should be eliminated. All the values are extremely small! Please check.
5. In the section (3.1), please use professional terminology to interpret spectra.
Author Response
For research article
Response to Reviewer 1 Comments |
||||||||||||||||||||||||||||||||||||||||||||||||||||||||||||||||||||||||||||||||||||
1. Summary |
|
|
||||||||||||||||||||||||||||||||||||||||||||||||||||||||||||||||||||||||||||||||||
Thank you very much for taking the time to review this manuscript. Please find the detailed responses below and the corresponding revisions/corrections highlighted/in track changes in the re-submitted files. |
||||||||||||||||||||||||||||||||||||||||||||||||||||||||||||||||||||||||||||||||||||
2. Questions for General Evaluation |
Reviewer's Evaluation |
Response and Revisions |
||||||||||||||||||||||||||||||||||||||||||||||||||||||||||||||||||||||||||||||||||
Does the introduction provide sufficient background and include all relevant references? |
Can be improved |
We have given our corresponding response in the point-by-point and completed a revision letter. |
||||||||||||||||||||||||||||||||||||||||||||||||||||||||||||||||||||||||||||||||||
Are all the cited references relevant to the research? |
Can be improved |
|||||||||||||||||||||||||||||||||||||||||||||||||||||||||||||||||||||||||||||||||||
Is the research design appropriate? |
Must be improved |
|||||||||||||||||||||||||||||||||||||||||||||||||||||||||||||||||||||||||||||||||||
Are the methods adequately described? |
Must be improved |
|||||||||||||||||||||||||||||||||||||||||||||||||||||||||||||||||||||||||||||||||||
Are the results clearly presented? |
Can be improved |
|||||||||||||||||||||||||||||||||||||||||||||||||||||||||||||||||||||||||||||||||||
Are the conclusions supported by the results? |
Can be improved |
|||||||||||||||||||||||||||||||||||||||||||||||||||||||||||||||||||||||||||||||||||
Reviewer 1 Comments and Suggestions: The author's research is meaningful for ensuring the quality of beef. However, there are some issues in the calibration process. Firstly, establish a model using formal chemometrics, and then evaluate the model using calibrated statistical parameters. Finally, use external data to validate the model. Thank you very much for your comments and suggestions. We have given our corresponding response in the point-by-point and completed a revision letter. The specific modifications are as follows. 3. Point-by-point response to Comments and Suggestions for Authors |
||||||||||||||||||||||||||||||||||||||||||||||||||||||||||||||||||||||||||||||||||||
Comments 1: Model transfer is meaningless for this study, please eliminate it. |
||||||||||||||||||||||||||||||||||||||||||||||||||||||||||||||||||||||||||||||||||||
Response 1: We thank for the reviewer's valuable suggestion. The main content of this study was to use near-infrared (NIR) hyperspectral spectroscopy to measure the TVC in stewed beef dishes samples in both intact and crushed physical states. Multiplicative Scattering Correction (MSC) was used to preprocess spectral data. And Competitive Adaptive Reweighted Sampling (CARS) was used to filter characteristic wavelengths. Kennard-Stone (KS) and Samples Set Partitioning Based on Joint X-Y Distance (SPXY)algorithms were used to select the optimal number of standard samples transferred by the model combined with root mean square error. The crushed samples were transferred into the complete samples prediction model through the Direct Standardization (DS) algorithm. The spectral hybrid model of crushed samples and full samples was established. Model transfer technology is an effective machine learning technology. Its essence is to use the output of one model as the input of another model. It can improve model performance and system efficiency, upgrade model quality, and reduce model training and prediction time. Model updating refers to updating and optimizing models in machine learning and artificial intelligence systems, which can improve the accuracy and generalization ability of the model and enable it to adapt better to new data and scenarios. In this study, the intact sample was less homogeneous than the crushed sample, resulting in more significant differences in spectral scattering and absorption coefficients. Thus, the accuracy and robustness of the colony count prediction model for the intact samples were not as good as that for the crushed sample. However crushed samples deprived the advantages of spectroscopy non-destructive testing. Therefore, we chose model transfer technology to transfer the prediction model of the crushed sample to the intact sample. The spectral hybrid model of the crushed sample and full sample was established, which can improve the utilization efficiency of the model and provide technical reference for real-time detection. |
||||||||||||||||||||||||||||||||||||||||||||||||||||||||||||||||||||||||||||||||||||
Comments 2: For the calibration, the main factors, R2cv, and RMSECV should be given, and they are very important parameters for the evaluation of models. |
||||||||||||||||||||||||||||||||||||||||||||||||||||||||||||||||||||||||||||||||||||
Response 2: Thanks for the reviewer's good question. The values of RMSECV and Rcv were added in the calibration. Leave-one-out cross-validation method was used to perform cross-validation. Only one remained as the test set at a time, while all others were used as the training set. To ensure consistency of the manuscript, Rcv instead of R2cv was used. “Root Mean Square Error of Cross-validation (RMSECV)” were added in Line 188 and189, the values of Rcv and RMSECV in Table 2 and Table 4, and highlighted them in the revised manuscript. Table 2. Calibration and prediction results of the TVC values of beef dishes for two physical morphology by NIR hyperspectral.
Table 4. Calibration and prediction results of the TVC values of beef dishes for two physical morphology by NIR hyperspectral under the new set of variables.
|
||||||||||||||||||||||||||||||||||||||||||||||||||||||||||||||||||||||||||||||||||||
Comments 3: There are many over-fittings for the calibration, e. g. Table 4 (CS), Table 5, Table 6, RMSEPs were much larger than RMSECs. Response 3: Thank you very much for your proposal. The values of RMSEP were larger than those of RMSEC in our research. One possible reason was that a small sample data set might cause the model to over-fitting. Saha et al. used NIR hyperspectral imaging to couple with chemometrics for rapid and non-destructive prediction of protein content in a single chickpea seed. Some of these results also showed that the values of RMSEP was larger than those of RMSEC. The other reason was that some key wavelengths containing helpful information might lose when using CARS. However, the RPD value of the model using CARS was quite high, indicating the model was a robust model. He et al. used NIR hyperspectral imaging combined with chemometric analysis to quickly and real-time predict lactic acid bacteria in farmed salmon flesh. In the spectral analysis based on full wavelengths, the RMSEP value of the LS-SVM, CARS-LS-SVM, and CARS-MLR models were all greater than the RMSEC. The above research showed that when using NIR hyperspectral to build a prediction model, the values of RMSEP may be larger than those of RMSEC. Tao et al. studied hyperspectral scattering characteristics to predict total viable count in beef. The research showed that different individual Lorentzian parameters and multivariate methods affect the model's accuracy. And among the best modeling results, the values of RMSEP from both PCR and PLSR methods was greater than those of RMSEC. In our study, the samples under different physical states were mainly analyzed and model transfer was used to achieve rapid and non-destructive detection of the total viable count in intact beef dishes, which could improve model reliability. After establishing the hybrid model, the values of RMSEP decreased by 9.23% more than before, and the values of RPD increased by 12.12%, respectively. The results showed that the stability and reliability of the HM model based on the CS and FS spectrum were improved after transmission. Regarding of the over-fittings for the calibration, we will could specialized research in the future to solve this problem by increasing the sample size. Comments 4: Table 1, for the unit (1g CFU/g), "1g" should be eliminated. All the values are extremely small! Please check. Response 4: We thank you for the reviewer's careful reading and good suggestion. The number of bacteria in the sample was log-transformed and expressed in 1g CFU/g for further analysis. To avoid objections, we replaced all 1g CFU/g with log10 CFU/g in Table 1, and line 511, respectively, and highlighted them in the revised manuscript. Comments 5: In the section (3.1), please use professional terminology to interpret spectra. Response 5: We thank you for the reviewer's good question. We have made some modifications to the content of section (3.1). We have replaced "Obvious troughs or peaks were observed near 944 nm, 980 nm, 1076 nm, 1186 nm, 1264 nm, and 1455 nm, indicating that there was a lot of frequency doubling and merging regions of chemical bonds in the samples, which formed absorption bands, and reflecting the difference of information content of different components [42]. The N-H stretching vibration has a second-order frequency doubling near 1076 nm. There was a tripling band of the C-H group near 1186 nm, which can obtain protein-related information. 1264 nm was the stretching vibration absorption band of the C-H group, which was related to fat content. There was a second-order frequency doubling absorption band of the O-H group near 950 nm and an O-H stretching first-order frequency doubling absorption band near 1455 nm, related to water molecules [43]. It can be seen from Figure 3 (b) that the peak shape of FS was consistent as a whole, with a few anomalies, which may be related to the uneven surface tissue of samples." with "The bands had many obvious peaks and valleys in the near 944 nm, 980 nm, 1076 nm, 1186 nm, 1264 nm, and 1455 nm. These absorption bands were formed by the overtone and combination vibrations of the molecular chemical bonds in the samples. The spectral data were contained a lot of information about the content of different components. [42]. The N-H stretching vibration has a second overtone near 1076 nm. A third overtone ab-sorption band of the C-H vibration near 1186 nm could obtain protein related information. 1264 nm was the stretching vibration absorption band of the C-H group, which was related to fat content. The water molecules may produce a peak at approximately 950 nm, which is associated with a second overtone frequency doubling absorption band of the O-H group. Another broad peak was located at approximately 1455 nm, possibly associated with the first overtone of the O-H stretching. [43]. It can be seen from Figure 4 (b) that the peak shape of FS was consistent overall, with a few anomalies, which may be related to the uneven surface tissue of samples." in line 302-314, and highlighted them in the revised manuscript. In line 339, we have added "spectral", and highlighted it in the revised manuscript. 4. Response to Comments on the Quality of English Language |
||||||||||||||||||||||||||||||||||||||||||||||||||||||||||||||||||||||||||||||||||||
Point 1: |
||||||||||||||||||||||||||||||||||||||||||||||||||||||||||||||||||||||||||||||||||||
Response 1: The reviewer didn't mention it. According to comments and suggestions, the grammar has been improved and some other minor modifications have been made in a careful proof reading. |
||||||||||||||||||||||||||||||||||||||||||||||||||||||||||||||||||||||||||||||||||||
5. Additional clarifications |
List of Changes (Red in the context)
- Table 2 and 4, which added RMSECV and Rcv values, have been inserted in the revised manuscript.
- The sentences “To”, “log10 CFU/g”, and “establishing” have been added on Page 1 of the revised manuscript.
- The sentences “spectrum's reflection, scattering, and absorption”, “could be used to”, “They”, and “We continued the team's previous work to solve the problem of poor accuracy and reproducibility of the complete sample prediction model” have been added on Page 2 of the revised manuscript.
- The sentences “CS and FS prediction model performance” and “The upper brains of cattle were boiled in cold water for 10 minutes to remove blood foam and stir-fried in a pot with hot oil” have been rewritten on Page 3 of the revised manuscript.
- The sentence “Finally, the number of bacteria in the sample was log-transformed, and expressed in log10 CFU/g for further analysis” was added on Page 4 of the revised manuscript.
- The words of “lg CFU/g” have been replaced with “log10 CFU/g” in Table 1. The sentence “As shown in Table 1”, “randomly” and “Root Mean Square Error of Cross-validation (RMSECV)” and “ɛ-SVR in the Lisvm toolbox was used in this study. The ɛ-SVR can achieve regression with strong robustness by introducing ɛ-insensitive loss function.” have been added on Page 5 of the revised manuscript.
- The sentences “3.1. Results of microbiological analyses” and “Figure. 3 depicts the reference the TVC values of beef dishes tested on each day of the experiment. During the storage period, the TVC value showed an upward trend and had certain significance. The TVC value reached the maximum value of 3.79 log10 CFU/g on the 10th day.” have been added on Page 7 of the revised manuscript.
- The “Figure 3” and the sentences “The bands had many obvious peaks and valleys in the near 944 nm, 980 nm, 1076 nm, 1186 nm, 1264 nm, and 1455 nm. These absorption bands were formed by the overtone and combination vibrations of the molecular chemical bonds in the samples. The spectral data were contained a lot of information about the content of different components. [42]. The N-H stretching vibration has a second overtone near 1076 nm. A third overtone absorption band of the C-H vibration near 1186 nm could obtain protein related information. 1264 nm was the stretching vibration absorption band of the C-H group, which was related to fat content. The water molecules may produce a peak at approximately 950 nm, which is associated with a second overtone frequency doubling absorption band of the O-H group. Another broad peak was located at approximately 1455 nm, possibly associated with the first overtone of the O-H stretching. [43]. It can be seen from Figure 4 (b) that the peak shape of FS was consistent overall, with a few anomalies, which may be related to the uneven surface tissue of samples.” and “spectral” have been added on Page 8 of the revised manuscript.
Figure 3. Reference beef dishes TVC values tested on each day.
- The sentences “Calibration and prediction results of the TVC values of beef dishes for two physical morphology by NIR hyperspectral.” and “The principle of the KS algorithm is to regard all samples as candidate samples of the calibration set, and select samples from them in order to enter the calibration set. It has the outstanding advantages of being fast, efficient, simple and intuitive, and highly representative when selecting samples. It can also ensure that the samples in the calibration set are evenly distributed according to spatial distance. The SPXY algorithm is developed based on the KS algorithm. It can take both x and y variables into account when calculating the distance between samples. And it can increase the difference and representativeness between samples, thereby reducing the number of samples in the calibration set and the amount of calculations in the modeling process.” were added on Page 10 of the revised manuscript.
- The sentence “Calibration and prediction results of the TVC values of beef dishes for two physical morphology by NIR hyperspectral under the new set of variables.” has been rewritten on Page 11 of the revised manuscript.
- The sentences of "log10 CFU/g" and “different SSNs' influence on the predicted results' RMSEP value“ have been added on Page 13 of the revised manuscript.
- The sentences of “log10 CFU/g” and “incremental transfer” have been added on Page 16 of the revised manuscript.

Reviewer 2 Report
Comments and Suggestions for Authors
From my understanding of the article, authors prepare 100 “whole samples”, analyze 10 every day by NIR, crush them, analyze them again and finally perform TVC on each. Thus, there is an expectation that the TVC increases as a function of time. If this is correct, then this information should be shown. If this is not true, the material and method section should be re-written to be clearer.
Since we are expecting an increase of TVC as a function of time, random selection of samples between calibration and prediction set is not ideal since not all the variability may be represented in each set. The addition of samples in Tables 7 and 8 should be performed as a function of the range in TVC and not randomly. Finally, the KS and SPXY selected samples should be representative of the range in TVC, is that the case?
Why use hyperspectral imaging if the spectra will all be averaged into 1 spectrum? Why not just collect a single point spectrum and use that? The only value of hyperspectral imaging is to identify local heterogeneity and use the information.
SVN parameters should be presented.
Lines 183-184 – “As shown in Table 1, the …”. Table 1 does not show that the spectra were pretreated by MSC
Equation 1. why not perform -log10 transfer of the data? It would help linearize the spectra
Table 2 and table 4 should have more descriptive titles. It is not clear just looking at them how the models are different.
Comments on the Quality of English LanguageThe article reads well but there are multiple awkward phrasings used across the manuscript that should be corrected.
Author Response
For research article
Response to Reviewer 2 Comments |
||
1. Summary |
|
|
Thank you very much for taking the time to review this manuscript. Please find the detailed responses below and the corresponding revisions/corrections highlighted/in track changes in the re-submitted files. |
||
2. Questions for General Evaluation |
Reviewer's Evaluation |
Response and Revisions |
Does the introduction provide sufficient background and include all relevant references? |
Can be improved |
We have given our corresponding response in the point-by-point. |
Are all the cited references relevant to the research? |
Yes |
|
Is the research design appropriate? |
Can be improved |
|
Are the methods adequately described? |
Can be improved |
|
Are the results clearly presented? |
Can be improved |
|
Are the conclusions supported by the results? |
Yes |
|
3. Point-by-point response to Comments and Suggestions for Authors Thank you very much for the reviewer's comments and suggestions. The reviewer didn't provide overall comments and suggestions for ours. We have given our corresponding response point-by-point. |
||
Comments 1: From my understanding of the article, authors prepare 100 "whole samples", analyze 10 every day by NIR, crush them, analyze them again and finally perform TVC on each. Thus, there is an expectation that the TVC increases as a function of time. If this is correct, then this information should be shown. If this is not true, the material and method section should be re-written to be clearer. |
||
Response 1: Thank you for the reviewer's suggestion and careful reading. The NIR hyperspectral spectra of the intact and crushed samples were separately collected. Within 10 days, each batch of samples was collected three times a day, and the average value was taken as the signal value. Analyzing NIR hyperspectral data and establishing the hybrid model aimed to improve the prediction accuracy of TVC in the intact samples. TVC was measured by plate colony counting. According to actual measurement results, the TVC values increased with the storage time of beef dishes. The sentences of “Figure. 3 Reference beef dishes TVC values tested on each day” and “Figure. 3 depicts the reference the TVC values of beef dishes tested on each day of the experiment. During the storage period, the TVC value showed an upward trend and had certain significance. The TVC value reached the maximum value of 3.79 log10 CFU/g on the 10th day.” have been added in the revised manuscript.
Figure. 3 Reference beef dishes TVC values tested on each day. |
||
Comments 2: Since we are expecting an increase of TVC as a function of time, random selection of samples between calibration and prediction set is not ideal since not all the variability may be represented in each set. The addition of samples in Tables 7 and 8 should be performed as a function of the range in TVC and not randomly. Finally, the KS and SPXY selected samples should be representative of the range in TVC, is that the case? |
||
Response 2: Thank you for the reviewer's good question. The principle of the KS algorithm is to regard all samples as candidate samples of the calibration set, and select samples from them to enter the calibration set. It has the outstanding advantages of being fast, efficient, simple, intuitive, and highly representative when selecting samples. It can also ensure that the samples in the calibration set are evenly distributed according to spatial distance. The SPXY algorithm is developed based on the KS algorithm. It can consider both x and y variables when calculating the distance between samples. And it can increase the difference and representativeness between samples, thereby reducing the number of samples in the calibration set and the amount of calculations in the modeling process. To improve the prediction performance of complete samples, the KS and SPXY algorithms were used for model transfer, respectively, and the hybrid model of the two physical forms was established. The results showed that the stability and reliability of the hybrid model established based on the spectrum of the crushed sample and the intact sample after transfer was enhanced. Meanwhile, the sentences of “The principle of the KS algorithm is to regard all samples as candidate samples of the calibration set, and select samples from them in order to enter the calibration set. It has the outstanding advantages of being fast, efficient, simple and intuitive, and highly representative when selecting samples. It can also ensure that the samples in the calibration set are evenly distributed according to spatial distance. The SPXY algorithm is developed based on the KS algorithm. It can take both x and y variables into account when calculating the distance between samples. And it can increase the difference and representativeness between samples, thereby reducing the number of samples in the calibration set and the amount of calculations in the modeling process.” have been added to Line 368-376, and highlighted in the revised manuscript. |
||
Comments 3: Why use hyperspectral imaging if the spectra will all be averaged into 1 spectrum? Why not just collect a single point spectrum and use that? The only value of hyperspectral imaging is to identify local heterogeneity and use the information. Response 3: We thank you for the reviewer's good question. The spectra varied from the solid medium's locations due to the inhomogeneous distribution of chemicals and tissue in intact beef dishes. The region of interest (ROI) was selected with hyperspectral imaging. ROI has richer information than a single point spectrum. ROI was chosen to focus on addressing the problem of spectral heterogeneity in the medium. Wang et al. and Jin et al. both extracted average spectral data from the ROI when using NIR hyperspectral imaging. Therefore, the ROI was averaged to a single spectrum, reducing errors. Comments 4: SVN parameters should be presented. Response 4: We thank you for the reviewer's good question. But we didn't use the SVN. In this research, SVM was used to establish a prediction model. By adjusting parameters such as kernel function, C parameter, and gamma parameter, SVM could effectively solve problems such as small samples, high dimensions, and nonlinearity. ɛ-SVR in the Lisvm toolbox was used in this study. The ɛ-SVR could achieve regression with strong robustness by introducing ɛ-insensitive loss function. Meanwhile, the sentences of “ɛ-SVR in the Lisvm toolbox was used in this study. The ɛ-SVR can achieve regression with strong robustness by introducing ɛ-insensitive loss function.” were added to Line 191-193, and highlighted them in the revised manuscript. Comments 5: Lines 183-184 – "As shown in Table 1, the …". Table 1 does not show that the spectra were pretreated by MSC. Response 5: We thank you for the reviewer's good suggestion, careful reading, and sorry for my wrong writing. Accordingly, we have rewritten the sentences of “As shown in Table 1, the 100 samples were randomly divided into a calibration and predic-tion set in the ratio of 4:1. The original spectra were pretreated by the MSC method” instead of “The 100 samples were divided into a calibration and prediction set in the ratio of 4:1 randomly. As shown in Table 1, the original spectra were pretreated by the MSC method” in Line 184-185, and highlighted them in the revised manuscript. Comments 6: Equation 1. why not perform -log10 transfer of the data? It would help linearize the spectra. Response 6: We thank you for the reviewer's good question and careful reading. The number of bacteria was log-transformed in our study and expressed in lg CFU/g. To avoid objections, we replaced all 1g CFU/g with log10 CFU/g in Table 1 and line 507, respectively. We have described the question in Line 146-148, and highlighted them in the revised manuscript. Comments 7: Table 2 and table 4 should have more descriptive titles. It is not clear just looking at them how the models are different. Response 7: We thank you for the reviewer's good suggestion and careful reading. Table 2 illustrates the comparison results of TVC content prediction models established using the same pretreatment and characteristic variable selection method for the two physical morphology spectra. Table 4 shows the comparison results of TVC content prediction models established using the new set of variables for the two physical morphology spectra. Accordingly, the sentences “Calibration and prediction results of the TVC values of beef dishes for two physical morphology by NIR hyperspectral” and “Calibration and prediction results of the TVC values of beef dishes for two physical morphology by NIR hyperspectral under the new set of variables” were rewritten on Table 2 and Table 4, respectively, and highlighted them in the revised manuscript. 4. Response to Comments on the Quality of English Language |
||
Point 1: Thank you very much for your comments and suggestions. The article reads well but there are multiple awkward phrasings used across the manuscript that should be corrected. |
||
Response 1: The grammar has been improved, and some other minor modifications have been made in a careful proofreading. |
||
5. Additional clarifications |
List of Changes (Red in the context)
- Table 2 and 4, which added RMSECV and Rcv values, have been inserted in the revised manuscript.
- The sentences “To”, “log10 CFU/g”, and “establishing” have been added on Page 1 of the revised manuscript.
- The sentences “spectrum's reflection, scattering, and absorption”, “could be used to”, “They”, and “We continued the team's previous work to solve the problem of poor accuracy and reproducibility of the complete sample prediction model” have been added on Page 2 of the revised manuscript.
- The sentences “CS and FS prediction model performance” and “The upper brains of cattle were boiled in cold water for 10 minutes to remove blood foam and stir-fried in a pot with hot oil” have been rewritten on Page 3 of the revised manuscript.
- The sentence “Finally, the number of bacteria in the sample was log-transformed, and expressed in log10 CFU/g for further analysis” was added on Page 4 of the revised manuscript.
- The words of “lg CFU/g” have been replaced with “log10 CFU/g” in Table 1. The sentence “As shown in Table 1”, “randomly” and “Root Mean Square Error of Cross-validation (RMSECV)” and “ɛ-SVR in the Lisvm toolbox was used in this study. The ɛ-SVR can achieve regression with strong robustness by introducing ɛ-insensitive loss function.” have been added on Page 5 of the revised manuscript.
- The sentences “3.1. Results of microbiological analyses” and “Figure. 3 depicts the reference the TVC values of beef dishes tested on each day of the experiment. During the storage period, the TVC value showed an upward trend and had certain significance. The TVC value reached the maximum value of 3.79 log10 CFU/g on the 10th day.” have been added on Page 7 of the revised manuscript.
- The “Figure 3” and the sentences “The bands had many obvious peaks and valleys in the near 944 nm, 980 nm, 1076 nm, 1186 nm, 1264 nm, and 1455 nm. These absorption bands were formed by the overtone and combination vibrations of the molecular chemical bonds in the samples. The spectral data were contained a lot of information about the content of different components. [42]. The N-H stretching vibration has a second overtone near 1076 nm. A third overtone absorption band of the C-H vibration near 1186 nm could obtain protein related information. 1264 nm was the stretching vibration absorption band of the C-H group, which was related to fat content. The water molecules may produce a peak at approximately 950 nm, which is associated with a second overtone frequency doubling absorption band of the O-H group. Another broad peak was located at approximately 1455 nm, possibly associated with the first overtone of the O-H stretching. [43]. It can be seen from Figure 4 (b) that the peak shape of FS was consistent overall, with a few anomalies, which may be related to the uneven surface tissue of samples.” and “spectral” have been added on Page 8 of the revised manuscript.
Figure 3. Reference beef dishes TVC values tested on each day.
- The sentences “Calibration and prediction results of the TVC values of beef dishes for two physical morphology by NIR hyperspectral.” and “The principle of the KS algorithm is to regard all samples as candidate samples of the calibration set, and select samples from them in order to enter the calibration set. It has the outstanding advantages of being fast, efficient, simple and intuitive, and highly representative when selecting samples. It can also ensure that the samples in the calibration set are evenly distributed according to spatial distance. The SPXY algorithm is developed based on the KS algorithm. It can take both x and y variables into account when calculating the distance between samples. And it can increase the difference and representativeness between samples, thereby reducing the number of samples in the calibration set and the amount of calculations in the modeling process.” were added on Page 10 of the revised manuscript.
- The sentence “Calibration and prediction results of the TVC values of beef dishes for two physical morphology by NIR hyperspectral under the new set of variables.” has been rewritten on Page 11 of the revised manuscript.
- The sentences of "log10 CFU/g" and “different SSNs' influence on the predicted results' RMSEP value“ have been added on Page 13 of the revised manuscript.
- The sentences of “log10 CFU/g” and “incremental transfer” have been added on Page 16 of the revised manuscript.

Round 2
Reviewer 1 Report
Comments and Suggestions for Authors
The quality of the manuscript has been improved, only some minor issues should be addressed.
The structure of “section 2.3.1 Preprocessing method of original spectral data” is chaotic and unclear
1. Table 1 is the results, not the method.
2. Line 188-192 is the method of model evaluation, not the preprocessing method. The model evaluation should be another section in the materials and methods.
3. Regarding the model evaluation, only the R2(C and CV) and RMSE (C and CV) can be used, as well as RPDcv. R2P and RMSEP were used to validate the model performance using external data.
4. Another important data LVs (or factor number) should be given.
Author Response
For research article
Response to Reviewer 1 Comments |
||||||||||||||||||||||||||||||||||||||||||||||||||||||||||||||||||||||||||||||||||||||||||||||||||||||||||||||||||||||||||||||||||||||||||||||||||||||||||||||||||||||||||||||||||||||||||||||||||||||||||||||||||||||||||||||||||||||||||||||||||||||||||||||||||||||||||||||||||||||||||||||||||||||||||||||||||||||||||||||||||||||||||||||||||||||||||||||||||||||||||||||||||||||||||||||||||||||||||||||||||||||||||||||||||||||||||||||||||||||||||||||||||||||||||||||||||||
1. Summary |
|
|
||||||||||||||||||||||||||||||||||||||||||||||||||||||||||||||||||||||||||||||||||||||||||||||||||||||||||||||||||||||||||||||||||||||||||||||||||||||||||||||||||||||||||||||||||||||||||||||||||||||||||||||||||||||||||||||||||||||||||||||||||||||||||||||||||||||||||||||||||||||||||||||||||||||||||||||||||||||||||||||||||||||||||||||||||||||||||||||||||||||||||||||||||||||||||||||||||||||||||||||||||||||||||||||||||||||||||||||||||||||||||||||||||||||||||||||||||
Thank you very much for taking the time to review this manuscript. Please find the detailed responses below and the corresponding revisions/corrections highlighted/in track changes in the re-submitted files. |
||||||||||||||||||||||||||||||||||||||||||||||||||||||||||||||||||||||||||||||||||||||||||||||||||||||||||||||||||||||||||||||||||||||||||||||||||||||||||||||||||||||||||||||||||||||||||||||||||||||||||||||||||||||||||||||||||||||||||||||||||||||||||||||||||||||||||||||||||||||||||||||||||||||||||||||||||||||||||||||||||||||||||||||||||||||||||||||||||||||||||||||||||||||||||||||||||||||||||||||||||||||||||||||||||||||||||||||||||||||||||||||||||||||||||||||||||||
2. Questions for General Evaluation |
Reviewer's Evaluation |
Response and Revisions |
||||||||||||||||||||||||||||||||||||||||||||||||||||||||||||||||||||||||||||||||||||||||||||||||||||||||||||||||||||||||||||||||||||||||||||||||||||||||||||||||||||||||||||||||||||||||||||||||||||||||||||||||||||||||||||||||||||||||||||||||||||||||||||||||||||||||||||||||||||||||||||||||||||||||||||||||||||||||||||||||||||||||||||||||||||||||||||||||||||||||||||||||||||||||||||||||||||||||||||||||||||||||||||||||||||||||||||||||||||||||||||||||||||||||||||||||||
Does the introduction provide sufficient background and include all relevant references? |
Can be improved |
We have given our corresponding response point-by-point and completed a revision letter. |
||||||||||||||||||||||||||||||||||||||||||||||||||||||||||||||||||||||||||||||||||||||||||||||||||||||||||||||||||||||||||||||||||||||||||||||||||||||||||||||||||||||||||||||||||||||||||||||||||||||||||||||||||||||||||||||||||||||||||||||||||||||||||||||||||||||||||||||||||||||||||||||||||||||||||||||||||||||||||||||||||||||||||||||||||||||||||||||||||||||||||||||||||||||||||||||||||||||||||||||||||||||||||||||||||||||||||||||||||||||||||||||||||||||||||||||||||
Are all the cited references relevant to the research? |
Can be improved |
|||||||||||||||||||||||||||||||||||||||||||||||||||||||||||||||||||||||||||||||||||||||||||||||||||||||||||||||||||||||||||||||||||||||||||||||||||||||||||||||||||||||||||||||||||||||||||||||||||||||||||||||||||||||||||||||||||||||||||||||||||||||||||||||||||||||||||||||||||||||||||||||||||||||||||||||||||||||||||||||||||||||||||||||||||||||||||||||||||||||||||||||||||||||||||||||||||||||||||||||||||||||||||||||||||||||||||||||||||||||||||||||||||||||||||||||||||
Is the research design appropriate? |
Can be improved |
|||||||||||||||||||||||||||||||||||||||||||||||||||||||||||||||||||||||||||||||||||||||||||||||||||||||||||||||||||||||||||||||||||||||||||||||||||||||||||||||||||||||||||||||||||||||||||||||||||||||||||||||||||||||||||||||||||||||||||||||||||||||||||||||||||||||||||||||||||||||||||||||||||||||||||||||||||||||||||||||||||||||||||||||||||||||||||||||||||||||||||||||||||||||||||||||||||||||||||||||||||||||||||||||||||||||||||||||||||||||||||||||||||||||||||||||||||
Are the methods adequately described? |
Can be improved |
|||||||||||||||||||||||||||||||||||||||||||||||||||||||||||||||||||||||||||||||||||||||||||||||||||||||||||||||||||||||||||||||||||||||||||||||||||||||||||||||||||||||||||||||||||||||||||||||||||||||||||||||||||||||||||||||||||||||||||||||||||||||||||||||||||||||||||||||||||||||||||||||||||||||||||||||||||||||||||||||||||||||||||||||||||||||||||||||||||||||||||||||||||||||||||||||||||||||||||||||||||||||||||||||||||||||||||||||||||||||||||||||||||||||||||||||||||
Are the results clearly presented? |
Must be improved |
|||||||||||||||||||||||||||||||||||||||||||||||||||||||||||||||||||||||||||||||||||||||||||||||||||||||||||||||||||||||||||||||||||||||||||||||||||||||||||||||||||||||||||||||||||||||||||||||||||||||||||||||||||||||||||||||||||||||||||||||||||||||||||||||||||||||||||||||||||||||||||||||||||||||||||||||||||||||||||||||||||||||||||||||||||||||||||||||||||||||||||||||||||||||||||||||||||||||||||||||||||||||||||||||||||||||||||||||||||||||||||||||||||||||||||||||||||
Are the conclusions supported by the results? |
Can be improved |
|||||||||||||||||||||||||||||||||||||||||||||||||||||||||||||||||||||||||||||||||||||||||||||||||||||||||||||||||||||||||||||||||||||||||||||||||||||||||||||||||||||||||||||||||||||||||||||||||||||||||||||||||||||||||||||||||||||||||||||||||||||||||||||||||||||||||||||||||||||||||||||||||||||||||||||||||||||||||||||||||||||||||||||||||||||||||||||||||||||||||||||||||||||||||||||||||||||||||||||||||||||||||||||||||||||||||||||||||||||||||||||||||||||||||||||||||||
Reviewer 1 Comments and Suggestions: The quality of the manuscript has been improved, only some minor issues should be addressed. The structure of "section 2.3.1 Preprocessing method of original spectral data" is chaotic and unclear. Thank you very much for your comments and suggestions. We have given our corresponding response in the point-by-point and completed a revision letter. The specific modifications are as follows. 3. Point-by-point response to Comments and Suggestions for Authors |
||||||||||||||||||||||||||||||||||||||||||||||||||||||||||||||||||||||||||||||||||||||||||||||||||||||||||||||||||||||||||||||||||||||||||||||||||||||||||||||||||||||||||||||||||||||||||||||||||||||||||||||||||||||||||||||||||||||||||||||||||||||||||||||||||||||||||||||||||||||||||||||||||||||||||||||||||||||||||||||||||||||||||||||||||||||||||||||||||||||||||||||||||||||||||||||||||||||||||||||||||||||||||||||||||||||||||||||||||||||||||||||||||||||||||||||||||||
Comments 1: Table 1 is the results, not the method. |
||||||||||||||||||||||||||||||||||||||||||||||||||||||||||||||||||||||||||||||||||||||||||||||||||||||||||||||||||||||||||||||||||||||||||||||||||||||||||||||||||||||||||||||||||||||||||||||||||||||||||||||||||||||||||||||||||||||||||||||||||||||||||||||||||||||||||||||||||||||||||||||||||||||||||||||||||||||||||||||||||||||||||||||||||||||||||||||||||||||||||||||||||||||||||||||||||||||||||||||||||||||||||||||||||||||||||||||||||||||||||||||||||||||||||||||||||||
Response 1: Thanks for the reviewer's valuable suggestion. Table 1 was moved to the section 3.1 Results of microbiological analyses, and the content of Table 1 was analyzed. The sentences "Table1. Measurement results of beef dishes TVC values in the calibration and prediction sets." and "Before establishing the prediction model, the 100 samples were randomly divided into a calibration and prediction set in the ratio of 4:1. Statistics of measurement results of beef dishes TVC values in the calibration set and prediction set are shown in Table 1. No significant difference was found between the calibration and prediction set's average and standard deviation. Thus, the division of the sample was reasonable." This has been added on Page 8, and highlighted in the revised manuscript. |
||||||||||||||||||||||||||||||||||||||||||||||||||||||||||||||||||||||||||||||||||||||||||||||||||||||||||||||||||||||||||||||||||||||||||||||||||||||||||||||||||||||||||||||||||||||||||||||||||||||||||||||||||||||||||||||||||||||||||||||||||||||||||||||||||||||||||||||||||||||||||||||||||||||||||||||||||||||||||||||||||||||||||||||||||||||||||||||||||||||||||||||||||||||||||||||||||||||||||||||||||||||||||||||||||||||||||||||||||||||||||||||||||||||||||||||||||||
Comments 2: Line 188-192 is the method of model evaluation, not the preprocessing method. The model evaluation should be another section in the materials and methods. |
||||||||||||||||||||||||||||||||||||||||||||||||||||||||||||||||||||||||||||||||||||||||||||||||||||||||||||||||||||||||||||||||||||||||||||||||||||||||||||||||||||||||||||||||||||||||||||||||||||||||||||||||||||||||||||||||||||||||||||||||||||||||||||||||||||||||||||||||||||||||||||||||||||||||||||||||||||||||||||||||||||||||||||||||||||||||||||||||||||||||||||||||||||||||||||||||||||||||||||||||||||||||||||||||||||||||||||||||||||||||||||||||||||||||||||||||||||
Response 2: Thanks for the reviewer's good question. The structure of section 2.3.1 Preprocessing method of original spectral data has been rewritten. The content of section 2.3.1 has been described the preprocessing method of original spectral data and model establishment. The sentences "The original spectra were pretreated by the MSC method to reduce the effect of interference, such as instrument baseline shifts, sample surface inhomogeneities, and other factors [27]. To simplify the model and heighten the prediction accuracy of the model, the Competitive Adaptive Reweighted Sampling (CARS) was used to screen the characteristic wavelengths [28,29]. The prediction model was established by using SVM [30]. ɛ-SVR in the Lisvm toolbox was used in this study. The ɛ-SVR can achieve regression with strong robustness by introducing ɛ-insensitive loss function." have been rewritten on Page 5 and highlighted in the revised manuscript. The content of section 2.3.2 Model performance evaluation has been added on Page 5. We have replaced "2.3.2 Model transfer", "2.3.3 Selection of the transfer samples", "2.3.4 The establishment of a hybrid model" with "2.3.3 Model transfer", "2.3.4 Selection of the transfer samples", "2.3.5 The establishment of a hybrid model", and highlighted them in the revised manuscript. |
||||||||||||||||||||||||||||||||||||||||||||||||||||||||||||||||||||||||||||||||||||||||||||||||||||||||||||||||||||||||||||||||||||||||||||||||||||||||||||||||||||||||||||||||||||||||||||||||||||||||||||||||||||||||||||||||||||||||||||||||||||||||||||||||||||||||||||||||||||||||||||||||||||||||||||||||||||||||||||||||||||||||||||||||||||||||||||||||||||||||||||||||||||||||||||||||||||||||||||||||||||||||||||||||||||||||||||||||||||||||||||||||||||||||||||||||||||
Comments 3: Regarding the model evaluation, only the R2(C and CV) and RMSE (C and CV) can be used, as well as RPDcv. R2P and RMSEP were used to validate the model performance using external data. Response 3: Thank you for your careful reading and good suggestion. Rc, Rcv and Rp were replaced by determination coefficient of calibration (R2 C), cross-validation (R2 CV), and prediction (R2 P), respectively, and highlighted in the revised manuscript. The values of R2 C, R2 CV and R2 P were added in Table 2 and Table 4, and the values of R2 C and R2 P were added in Table 5, Table 6, Table 7 and Table 8, and highlighted in the revised manuscript. Table 2. Calibration and prediction results of the TVC values of beef dishes for two physical morphology by NIR hyperspectral.
Table 4. Calibration and prediction results of the TVC values of beef dishes for two physical morphology by NIR hyperspectral under the new set of variables.
Table 5. The modeling results of optimal SSN for FS were selected from the standard set of crushed correction samples.
Table 6. The modeling results of the optimal SSN for FS were selected with the full calibration sample as the standard set.
Table 7. CS and FS hybrid model predict the results of FS.
Table 8. The prediction results of FS using CS and transferred FS hybrid model.
Comments 4: Another important data LVs (or factor number) should be given. Response 4: Thank you for your proposal. Latent Variables (LVs) is an important parameter that affects model performance. Too many LVs will make over-fitting of the model, resulting in a decrease in the model's predictive ability. Too few LVs will lead to under-fitting of the model, resulting in insufficient explanatory power of the model. The best LVs was selected by the number of LVs corresponding to the minimum RESEP. It can be seen that compared with the previous modeling results in Table 2, R2 P of the FS prediction set increased from 0.3453 to 0.5088 in Table 4, and the values of LVs have decreased, but the results of RMSE and RPD were not ideal. RPD was still less than 2.5, and the reliability of the model was poor. Therefore, follow-up research was conducted and the hybrid model was established to improve the crushing model's ability to predict FS. The values of LVs were added in Table 2 and Table 4, respectively, and the sentences "Besides, Latent Variables (LVs) is also an important parameter that affects model performance." have been added in Lines 195-196, and highlighted in the revised manuscript. Table 2. Calibration and prediction results of the TVC values of beef dishes for two physical morphology by NIR hyperspectral.
Table 4. Calibration and prediction results of the TVC values of beef dishes for two physical morphology by NIR hyperspectral under the new set of variables.
4. Response to Comments on the Quality of English Language |
||||||||||||||||||||||||||||||||||||||||||||||||||||||||||||||||||||||||||||||||||||||||||||||||||||||||||||||||||||||||||||||||||||||||||||||||||||||||||||||||||||||||||||||||||||||||||||||||||||||||||||||||||||||||||||||||||||||||||||||||||||||||||||||||||||||||||||||||||||||||||||||||||||||||||||||||||||||||||||||||||||||||||||||||||||||||||||||||||||||||||||||||||||||||||||||||||||||||||||||||||||||||||||||||||||||||||||||||||||||||||||||||||||||||||||||||||||
Point 1: |
||||||||||||||||||||||||||||||||||||||||||||||||||||||||||||||||||||||||||||||||||||||||||||||||||||||||||||||||||||||||||||||||||||||||||||||||||||||||||||||||||||||||||||||||||||||||||||||||||||||||||||||||||||||||||||||||||||||||||||||||||||||||||||||||||||||||||||||||||||||||||||||||||||||||||||||||||||||||||||||||||||||||||||||||||||||||||||||||||||||||||||||||||||||||||||||||||||||||||||||||||||||||||||||||||||||||||||||||||||||||||||||||||||||||||||||||||||
Response 1: The reviewer didn't mention it. According to comments and suggestions, the grammar has been improved, and some other minor modifications have been made in a careful proofreading. |
||||||||||||||||||||||||||||||||||||||||||||||||||||||||||||||||||||||||||||||||||||||||||||||||||||||||||||||||||||||||||||||||||||||||||||||||||||||||||||||||||||||||||||||||||||||||||||||||||||||||||||||||||||||||||||||||||||||||||||||||||||||||||||||||||||||||||||||||||||||||||||||||||||||||||||||||||||||||||||||||||||||||||||||||||||||||||||||||||||||||||||||||||||||||||||||||||||||||||||||||||||||||||||||||||||||||||||||||||||||||||||||||||||||||||||||||||||
5. Additional clarifications |
List of Changes (Red in the context)
- Table 2 and 4, which added RMSECV, R2 cv, R2 C, R2 P and LVs values, have been inserted in the revised manuscript. And the values of R2 C and R2 P were added in Table 5, Table 6, Table 7 and Table 8, and highlighted them in the revised manuscript.
- The sentences "To", "Determination Coefficient of Calibration (R2 P)", "0.5088 to 0.8068", "log10 CFU/g", "establishing", "food hygiene quality", "many bacteria could grow in the storage process" and "greatly threaten" have been added on Page 1 of the revised manuscript.
- The sentences "spectrum's reflection, scattering, and absorption", "could be used to", "They", and "We continued the team's previous work to solve the problem of poor accuracy and reproducibility of the complete sample prediction model" have been added on Page 2 of the revised manuscript.
- The sentences "CS and FS prediction model performance" and "The upper brains of cattle were boiled in cold water for 10 minutes to remove blood foam and stir-fried in a pot with hot oil" have been rewritten on Page 3 of the revised manuscript.
- The sentence "Finally, the number of bacteria in the sample was log-transformed, and expressed in log10 CFU/g for further analysis" was added on Page 4 of the revised manuscript.
- The sentences "and model establishment", "to reduce the effect of interference, such as instrument baseline shifts, sample surface in-homogeneities, and other factors [27]. To simplify the model and heighten the prediction accuracy of the model, the Competitive Adaptive Reweighted Sampling (CARS) was used to screen the characteristic wavelengths” "ɛ-SVR in the Lisvm toolbox was used in this study. The ɛ-SVR can achieve regression with strong robustness by introducing ɛ-insensitive loss function.", "2.3.2 Model performance evaluation" and "Determination Coefficient of Calibration (R2 C), Cross-validation (R2 CV), and Prediction (R2 P) , Root Mean Squared Error of Calibration (RESEC), Root Mean Square Error of Cross-validation (RMSECV)", "performance", and "Besides, Latent Variables (LVs) is also an important parameter that affects model performance.", have been added on Page 5 of the revised manuscript.
- The sentences "3.1. Results of microbiological analyses" and "Figure. 3 depicts the reference the TVC values of beef dishes tested on each day of the experiment. During the storage period, the TVC value showed an upward trend and had certain significance. The TVC value reached the maximum value of 3.79 log10 CFU/g on the 10th day." have been added on Page 7 of the revised manuscript.
- The "Figure 3", "Table 1" and the sentences "Before establishing the prediction model, the 100 samples were randomly divided into a calibration and prediction set in the ratio of 4:1. Statistics of measurement results of beef dishes TVC values in the calibration set and prediction set are shown in Table 1. No significant difference was found between the calibration and prediction set's average and standard deviation. Thus, the samples division was reasonable.", and "The bands had many obvious peaks and valleys in the near 944 nm, 980 nm, 1076 nm, 1186 nm, 1264 nm, and 1455 nm. These absorption bands were formed by the overtone and combination vibrations of the molecular chemical bonds in the samples. The spectral data were contained a lot of information about the content of different components. [42]. The N-H stretching vibration has a second overtone near 1076 nm. A third overtone absorption band of the C-H vibration near 1186 nm could obtain protein-related information. 1264 nm was the stretching vibration absorption band of the C-H group, which was related to fat content. The water molecules may produce a peak at approximately 950 nm, which is associated with a second overtone frequency doubling absorption band of the O-H group. Another broad peak was located at approximately 1455 nm, possibly associated with the first overtone of the O-H stretching. [43]. It can be seen from Figure 4 (b) that the peak shape of FS was consistent overall, with a few anomalies, which may be related to the uneven surface tissue of samples." This has been added on Page 8 of the revised manuscript.
Figure 3. Reference beef dishes TVC values tested on each day.
Table1. Measurement results of beef dishes TVC values in the calibration and prediction sets.
Samples type |
Samples size |
The maximum/ (log10 CFU/g) |
The minimum/ (log10 CFU/g) |
The average/ (log10 CFU/g) |
Standard deviation/ (log10 CFU/g) |
Calibration set |
80 |
3.8808 |
2.1347 |
3.1140 |
0.2953 |
Prediction set |
20 |
3.7559 |
2.1047 |
3.1490 |
0.3623 |
- The words of "spectral" have been edded on Page 9 of the revised manuscript.
- The sentences "Calibration and prediction results of the TVC values of beef dishes for two physical morphology by NIR hyperspectral." and "The principle of the KS algorithm is to regard all samples as candidate samples of the calibration set, and select samples from them in order to enter the calibration set. It has the outstanding advantages of being fast, efficient, simple, intuitive, and highly representative when selecting samples. It can also ensure that the samples in the calibration set are evenly distributed according to spatial distance. The SPXY algorithm is developed based on the KS algorithm. It can consider both x and y variables when calculating the distance between samples. And it can increase the difference and representativeness between samples, thereby reducing the number of samples in the calibration set and the amount of calculations in the modeling process." Page 10 of the revised manuscript added.
- The sentence "R2 P of the FS prediction set increased from 0.3453 to 0.5088", and "Calibration and prediction results of the TVC values of beef dishes for two physical morphology by NIR hyperspectral under the new set of variables." have been rewritten on Page 12 of the revised manuscript.
- The sentences " the value of R2 P was 0.8069, the value of RMSEP was 0.1691 log10 CFU/g" and "different SSNs' influence on the predicted results' RMSEP value "have been added on Page 13 of the revised manuscript.
- The word "R2 P" has been added on Page 13 of the revised manuscript.
- The sentences "R2 P=0.8069", "log10 CFU/g" and "incremental transfer" have been added on Page 16 of the revised manuscript.
